# Angiotensin-converting enzyme polymorphisms AND Alzheimer's disease susceptibility: An updated meta-analysis

**Xiao-Yu Xin** **, Ze-Hua Lai, Kai-Qi Ding, Li-Li Zeng\*, Jian-Fang Ma\***

Department of Neurology and Institute of Neurology, Ruijin Hospital, School of Medicine, Shanghai Jiaotong University, Shanghai, China

\* llzeng@126.com (LLZ); mjf10924@rjh.com.cn (JFM)

## Abstract

### Background

Many studies among different ethnic populations suggested that angiotensin converting enzyme (ACE) gene polymorphisms were associated with susceptibility to Alzheimer's disease (AD). However, the results remained inconclusive. In the present meta-analysis, we aimed to clarify the effect of ACE polymorphisms on AD risk using all available relevant data.

### Methods

Systemic literature searches were performed using PubMed, Embase, Alzgene and China National Knowledge Infrastructure (CNKI). Relevant data were abstracted according to pre-defined criteria.

### Results

Totally, 82 independent cohorts from 65 studies were included, focusing on five candidate polymorphisms. For rs1799752 polymorphism, in overall analyses, the insertion (*I*) allele conferred increased risk to AD compared to the deletion (*D*) allele (*I* vs. *D*: OR = 1.091, 95% CI = 1.007–1.181, *p* = 0.032); while the *I* carriers showed increased AD susceptibility compared with the *D* homozygotes (*II* + *ID* vs. *DD*: OR = 1.131, 95% CI = 1.008–1.270, *p* = 0.036). However, none of the positive results passed FDR adjustment. In subgroup analysis restricted to late-onset individuals, the associations between rs1799752 polymorphism and AD risk were identified using allelic comparison (OR = 1.154, 95% CI = 1.028–1.295, *p* = 0.015, FDR = 0.020), homozygotes comparison, dominant model and recessive model (*II* vs. *ID* + *DD*: OR = 1.272, 95% CI = 1.120–1.444, *p* < 0.001, FDR < 0.001). Nevertheless, no significant association could be revealed after excluding studies not in accordance with Hardy-Weinberg equilibrium (HWE). In North Europeans, but not in East Asians, the *I* allele demonstrated increased AD susceptibility compared to the *D* allele (OR = 1.096, 95% CI = 1.021–1.178, *p* = 0.012, FDR = 0.039). After excluding HWE-deviated cohorts, significant associations were also revealed under homozygotes comparison, additive model (*ID* vs.

**Funding:** This study was supported by grants from the National Natural Science Foundation of China (81301046). The funders had no role in study design, data collection and analysis, decision to publish, or preparation of the manuscript.

**Competing interests:** No authors have competing interests.

*DD*: OR = 1.266, 95% CI = 1.045–1.534, *p* = 0.016, FDR = 0.024) and dominant model (*II* + *ID* vs. *DD*: OR = 1.197, 95% CI = 1.062–1.350, *p* = 0.003, FDR = 0.018) in North Europeans. With regard to rs1800764 polymorphism, significant associations were identified particularly in subgroup of European descent under allelic comparison (*T* vs. *C*: OR = 1.063, 95% CI = 1.008–1.120, *p* = 0.023, FDR = 0.046), additive model and dominant model (*TT* + *TC* vs. *CC*: OR = 1.116, 95% CI = 1.018–1.222, *p* = 0.019, FDR = 0.046). But after excluding studies not satisfying HWE, all these associations disappeared. No significant associations were detected for rs4343, rs4291 and rs4309 polymorphisms in any genetic model.

## Conclusions

Our results suggested the significant but modest associations between rs1799752 polymorphism and risk to AD in North Europeans. While rs4343, rs4291 and rs4309 polymorphisms are unlikely to be major factors in AD development in our research.

## Introduction

Alzheimer's disease (AD) is an insidious neurodegenerative disorder characterized by progressive cognitive decline, especially irreversible memory impairment. As the most common form of dementia worldwide, AD accounts for about 50~70% of all dementia cases. In China, it was estimated that 9,83 million people aged 60 years or older suffer from AD [1]. Although the precise mechanisms of pathogenesis have not yet been fully defined, with epidemiological and molecular evidence, AD is considered as a multifactorial disease attributed to a complex interaction of both genetic and environment factors. While heritable factors account for 60–80% of AD risk [2].

Despite a number of rare mutations on the Aβ precursor protein (APP), Presenilin-1 (PS1) and Presenilin-2 (PS2) relating to familial AD, which account for less than 2% of all AD cases, the apolipoprotein E (APOE) ε4 allele remains the strongest genetic risk factor for sporadic AD. Previous studies linked APOE with Aβ aggregation and clearance, tau neurofibrillary degeneration, microglia and astrocyte responses, and blood-brain barrier (BBB) disruption [3]. Presence of APOE ε4 allele increases risk of AD with a dose-dependent manner, and might lead to an earlier age of disease onset. The frequency of APOE ε4 in AD patients varied among different ethnic groups, ranging from around 40% to 60%, compared to 20%~25% in controls. Therefore, the presence of ε4 is neither necessary nor sufficient to cause the disease, indicating the participant of other heritable risk factors underlying the development of AD [4, 5].

Recently, many evidences supported that ACE participated in the pathogenesis of AD. As a membrane-bound zinc metalloprotease, ACE played an important role in Aβ degradation. Angiotensin converting enzyme (ACE) is an important component of the renin-angiotensin system (RAS), which mainly acts on promoting the formation of Angiotensin II (Ang II) from Angiotensin I (Ang I) [6]. In an 8-year longitudinal study, the mean intelligence quotient of male hypertensive patients taking ACE inhibitors declined more rapidly than that of others taking no ACE inhibitors. In human APP/ACE +/- mice, a decrease in ACE levels promoted $A\beta_{42}$ deposition and increased the number of apoptotic neurons [7]. Peripherally derived ACE-enhanced macrophage reduced cerebral soluble $A\beta_{42}$ level and alleviated vascular and parenchymal Aβ deposits [8]. All these results confirmed the role of ACE in AD development.

The ACE gene is located on chromosome 17q23. The most common polymorphism of ACE gene is the insertion/deletion (I/D) variant of 287-bp in intron 16 (rs1799752). The I/D

genotype is regarded as a determinant of ACE expression levels in plasma, cells and tissues. Approximately 50% variability in plasma levels of ACE depends on the rs1799752 polymorphism [9, 10]. Individuals carrying the *D* allele have higher plasma ACE levels compared to *I* homozygotes [11]. Moreover, rs1799752 I/D polymorphism has been reported to link with coronary heart disease and cognitive impairment, for example, AD and vascular dementia (VD). Besides rs1799752 I/D polymorphism, several other polymorphisms of the ACE gene were also investigated in AD cohorts of different ethnics, such as rs1800764 T/C, rs4343 A/G, rs4291 A/T polymorphisms, et al. rs1800764 and rs4291 located in the regulatory region of ACE gene, while rs4343 in the exotic region. Though some studies have demonstrated the associations, inconsistency was still presented among different study populations. These discrepancies may be related to the small sample size of individual studies, the difference in ethnic background and the different methodologies used for analysis. While meta-analysis is a well-established means to quantitatively synthesize all association data across studies to reduce heterogeneity and identify minor genetic effects, which largely addressing the issue of sample size.

Thus, in the present study, we performed an updated meta-analysis combining all available case-control studies to derive a more precise estimation of the associations between ACE gene polymorphisms and AD susceptibility. We also stratified the study cohorts, when possible, according to the age of onset and ethnic background. Since some recent evidence suggested that the presence of APOE $\varepsilon4$ influence the behavioural effects of ACE I/D polymorphism in AD, and the protective effects of ACE inhibitors or angiotensin receptor blockers on cognitive decline correlated with APOE $\varepsilon4$ carrier status, we also performed subgroup analyses according to APOE $\varepsilon4$ carrier status if sufficient data could be obtained [12, 13].

## Materials and methods

### Literature search

We performed computerized searches of PubMed, Embase, Alzgene and China National Knowledge Infrastructure (CNKI) up to January 31st, 2021. The following keywords were used: (angiotensin-converting enzyme or ACE or DCP1) AND (Alzheimer or dementia) AND (polymorphism or variant or allele or genotype), with no language restriction. In addition, references of retrieved articles, reviews and meta-analyses were checked manually for potential studies.

### Study selection

Studies included in the meta-analysis should meet the following criteria: (1) case-control design; (2) the evaluation of the relationship between the ACE polymorphisms and AD; (3) AD was diagnosed according to generally accepted criteria, such as criteria of the National Institute of Neurological and Communicative Disorders and Stroke–Alzheimer's Disease and Related Disorders Association (NINCDS-ADRDA) or of the Diagnostic and Statistical Manual of Mental Disorders-IV (DSM-IV), et al.; (4) genotype or allele frequencies were available in both cases and controls. Studies that performed in more than 1 population were considered as separate investigations. When the articles contained duplicated data, the most recent or complete data set was selected. Since all the included studies were gene polymorphism investigations in patients and controls, which were not suitable for randomized controlled design, no randomized controlled trial (RCT) could be identified and involved in our research.

### Data extraction

Data were extracted from the eligible articles by two investigators independently and agreements were achieved on all items. The following information was collected using a predefined

reporting form: name of the first author, publication year, sample size, country, racial descent, diagnosis criteria of AD, genotyping method, source of controls, distribution of allele and genotypes in both AD and control groups, and Hardy-Weinberg equilibrium (HWE) in controls. The Newcastle-Ottawa Scale (NOS) was performed to assess the methodological quality of the included studies. The total NOS score ranges from 0 to 9. If the score was 6 or more, the study was assumed to be high quality [14].

## Statistical analysis

The STATA Software (version 14.0, Stata Corp) was used for analytical procedures. Hardy–Weinberg equilibrium (HWE) was assessed in the controls using the exact test. For each gene polymorphism, meta-analysis was performed only if data were available from at least 4 independent studies. We did not assume a genetic model in advance. Firstly, odds ratio was produced by allelic comparison. Secondly, we compared each genotype with one other in turn. Thirdly, we compared each genotype in turn with the other two combined. All these three methods have been widely used for pooling data in genetic association studies [15].

The strength of association was determined by pooled odds ratio (OR) along with the corresponding 95% confidence interval (CI). The Dersimonian and Laird's Q test was performed for heterogeneity evaluation. If the $p$ value was less than 0.10, the heterogeneity was considered statistically significant. Quantification of the heterogeneity was assessed using the $I^2$ metric, which represents the percentage of the observed between study variabilities due to heterogeneity rather than due to chance. $I^2$ ranges from 0% to 100%, with higher values indicating a greater degree of heterogeneity. Where there was significant heterogeneity among studies ($I^2 \geq 50\%$), the pooled OR was calculated by a random-effect model (Dersimonian and Laird); Otherwise, a fixed-effect model was used (Mantel-Haenszel).

Stratified analysis was performed, when feasible, by geographic location, age of onset, or APOE $\varepsilon 4$ status. In addition, pooled odds ratios were calculated particularly in large sample size cohorts ($\geq 300$ participants). Cumulative meta-analysis was conducted to investigate the trend and the stability of risk effects as evidence accumulated over time. To adjust for multiple comparisons, we applied the Benjamini-Hochberg (BH) method to control false discovery rate (FDR) [16]. Publication bias was examined using both the Begg-Mazumdar test and the Egger's regression asymmetry test.

## Results

### Eligible studies and candidate polymorphisms

77 articles covered 35 polymorphisms were identified as potential candidates after primary electronic searches and manual screening. Among these, 9 articles were excluded for duplicated data. 2 articles were excluded since there was no sufficient number of studies for a meta-analysis (more than four separate studies were required for a meta-analysis in the present investigation) [17, 18]. For 1 instance, because exact allele or genotype counts could not be obtained despite attempts to contact the authors, it was also not involved [19]. Therefore, 65 articles with 82 samples were finally included in our study, focusing on 5 polymorphisms as following: rs1799752 I/D, rs1800764 T/C, rs4343 A/G, rs4291 A/T, rs4309 C/T (Table 1). The mean number of samples per candidate polymorphism was 24.80 ± 19.49 (Table 2). Main diagnostic criteria for AD included NINCDS-ADRDA, DSM-IV, ICD-10, CREAD. All the included studies used standard genotyping method in laboratory. Fig 1. shows the detailed screening process for the involved literature.

**Table 1. General characteristics of included studies in the present meta-analysis.**

| Author | Year | Country | Ethnicity | Diagnostic criteria | Sample size (cases/controls) | Polymorphisms | NOS[1] |
|---|---|---|---|---|---|---|---|
| Chapman [20] | 1998 | Israel | European descent | NINCDS-ADRDA[2], probable; DSM-III-R[3] | 49/40 | 1799752 | 8 |
| Scacchi [21] | 1998 | Italy | European descent | NINCDS-ADRDA, probable | 80/155 | 1799752 | 7 |
| Alveraz [22] | 1999 | Spain | European descent | NINCDS-ADRDA, probable | 350/517 | 1799752 | 8 |
| Hu [23] | 1999 | Japan | Japanese | NINCDS-ADRDA, probable | 132/257 | 1799752 | 7 |
| Kehoe (3 cohorts) [24] | 1999 | Ireland | European descent | NINCDS-ADRDA | 542/386 | 1799752 | 7 |
| Palumbo [25] | 1999 | Italy | European descent | NINCDS-ADRDA, probable, possible | 140/40 | 1799752 | 8 |
| Crawford [26] | 2000 | USA | Mixed | NINCDS-ADRDA, probable, possible | 171/175 | 1799752 | 9 |
| Farrer (2 cohorts) [27] | 2000 | USA; Canada; Italy; Russia | European descent | NINCDS-ADRDA | 386/375 | 1799752 | 8 |
| Mattila [28] | 2000 | Finland | European descent | NINCDS-ADRDA, probable; CREAD | 80/67 | 1799752 | 8 |
| Myllykangas [29] | 2000 | Finland | European descent | CERAD4, probable, definite | 121/75 | 1799752 | 8 |
| Narain [30] | 2000 | UK | European descent | CERAD | 239/342 | 1799752 | 7 |
| Yang [31] | 2000 | China | Chinese | NINCDS-ADRDA, probable; DSM-IV5 | 188/227 | 1799752 | 8 |
| Isbir [32] | 2001 | Turkey | European descent | NINCDS-ADRDA, probable | 35/29 | 1799752 | 7 |
| Perry [33] | 2001 | USA | African American | NINCDS-ADRDA, probable, definite; CERAD | 111/78 | 1799752 | 8 |
| Prince [34] | 2001 | Sweden | European descent | NINCDS-ADRDA, probable, definite; CERAD | 204/186 | 4343 | 8 |
| Richard (2 cohorts) [35] | 2001 | France | European descent | NINCDS-ADRDA, probable; DSM-III-R | 56/221 | 1799752 | 9 |
| Zuiliani [36] | 2001 | Italy | European descent | NINCDS-ADRDA, probable | 40/54 | 1799752 | 8 |
| Buss [37] | 2002 | Germany; Switzerland; Italy | European descent | NINCDS-ADRDA | 261/306 | 1799752 | 8 |
| Cheng [38] | 2002 | China | Chinese | NINCDS-ADRDA, probable | 173/285 | 1799752 | 7 |
| Lendon [39] | 2002 | UK | European descent | NINCDS-ADRDA, probable; DSM-III-R; | 214/99 | 1799752 | 7 |
| Monastero [40] | 2002 | Italy | European descent | NINCDS-ADRDA, probable | 149/149 | 1799752 | 9 |
| Panza [41] | 2002 | Italy | European descent | NINCDS-ADRDA, probable | 141/268 | 1799752 | 9 |
| Wu [42] | 2002 | China | Chinese | DSM-IV | 96/96 | 1799752 | 7 |
| Carbonell [43] | 2003 | UK | European descent | NINCDS-ADRDA possible probable | 80/65 | 1799752 | 7 |
| Kehoe (4 cohorts) [44] | 2003 | Sweden; UK | European descent | NINCDS-ADRDA, possible, probable, definite; CERAD | 846/773 | 1799752,4343, 4291,1800764 | 9 |
| Seripa (2 cohorts) [45] | 2003 | Italy; USA | European descent | NINCDS-ADRDA; probable | 250/203 | 1799752 | 9 |
| Camelo [46] | 2004 | Colombia | European descent | NINCDS-ADRDA; probable | 83/69 | 1799752 | 8 |
| Feng [47] | 2004 | China | Chinese | NINCDS-ADRDA | 26/68 | 1799752 | 7 |
| Koelsch [48] | 2005 | Germany | European descent | DSM-IV | 351/348 | 1799752 | 9 |

*(Continued)*

**Table 1.** (Continued)

| Author | Year | Country | Ethnicity | Diagnostic criteria | Sample size (cases/controls) | Polymorphisms | NOS[1] |
|---|---|---|---|---|---|---|---|
| Lehmann [49] | 2005 | UK | European descent | NINCDS-ADRDA; DSM-IV | 203/248 | 1799752 | 8 |
| Sleegers [50] | 2005 | Netherlands | European descent | NINCDS-ADRDA, probable; DSM-III-R | 250/6403 | 1799752 | 9 |
| Zhang [51] | 2005 | China | Chinese | NINCDS-ADRDA; | 192/195 | 1799752 | 8 |
| Blomsqvist [52] | 2006 | UK; Sweden | European descent | NINCDS-ADRDA, probable, definite; CERAD | 940/405 | 4309 | |
| Keikhaee [53] | 2006 | Iran | European descent | NINCDS-ADRDA, probable | 117/125 | 1799752 | 9 |
| Meng [54] | 2006 | Israel | European descent | DSM-IV | 92/166 | 1800764,4291,4343 | 7 |
| Wang [55] | 2006 | China | Chinese | NINCDS-ADRDA; DSM-III-R | 104/128 | 1799752 | 8 |
| Wang [56] | 2006 | China | Chinese | DSM-III-R, NINCDS-ADRDA, probable | 151/161 | 1799752 | 7 |
| Wehr [57] | 2006 | Poland | European descent | NINCDS-ADRDA | 100/144 | 1799752 | 7 |
| Liu [58] | 2007 | China | Chinese | NINCDS-ADRDA, probable; DSM-IV | 39/50 | 1799752 | 8 |
| Nacmias [59] | 2007 | Italy | European descent | DSM-IV | 388/303 | 1799752 | 9 |
| Bruandet [60] | 2008 | France | European descent | NINCDS-ADRDA; DSM-IV | 141/6467 | 4291,4343 | 8 |
| Han [61] | 2008 | China | Chinese | NINCDS-ADRDA, probable | 55/59 | 1799752 | 9 |
| Trebunova [62] | 2008 | Slovakia | European descent | NINCDS-ADRDA | 70/126 | 1799752 | 8 |
| Giedraitis [63] | 2009 | Sweden | European descent | DSM-IV, NINCDS-ADRDA | 86/404 | 4343 | 9 |
| Helbecque [64] | 2009 | France, UK, Spain, Netherlands, Italy | European descent | NINCDS-ADRDA; DSM-III-R | 376/444 | 4291,4343 | 7 |
| Vardy [65] | 2009 | UK | European descent | NINCDS-ADRDA | 94/188 | 1799752 | 8 |
| Corneveaux [66] | 2010 | UK, USA, Netherland | European descent | CERAD | 1019/591 | 1800764 | 9 |
| Feulner [67] | 2010 | Germany | European descent | NINCDS-ADRDA | 491/479 | 4309 | 8 |
| Ning [68] | 2010 | China | Chinese | NINCDS-ADRDA; DSM-IV | 144/476 | 1799752,4343, 1800764 | 8 |
| Sarajarvi [69] | 2010 | Finland | European descent | NINCDS-ADRDA, probable | 642/682 | 4343 | 7 |
| shulman [70] | 2010 | USA | European descent | CERAD | 173/131 | 1800764 | 9 |
| Belbin (10 cohorts) [71] | 2011 | UK | European descent | NINCDS-ADRDA; CERAD | 3930/4282 | 4291,4343,1800764 | 7 |
| Cousin [72] | 2011 | France | European descent | NINCDS-ADRDA, probable | 428/475 | 1799752,4291 | 8 |
| Ghebranious [73] | 2011 | USA | European descent | NINCDS-ADRDA | 153/302 | 4343,4291,1800764 | 7 |
| Lucatelli [74] | 2011 | Brazil | Mixed | DSM-IV, NINCDS-ADRDA | 35/85 | 1799752 | 8 |
| Nirmal [75] | 2011 | India | Indian | DSM-IV | 95/130 | 1799752 | 7 |
| Yang [76] | 2011 | China | Chinese | NINCDS-ADRDA, probable | 257/137 | 1799752 | 9 |
| Zhang [77] | 2014 | China | Chinese | NINCDS-ADRDA | 96/102 | 1799752 | 7 |
| Deng [78] | 2015 | China | Chinese | NINCDS-ADRDA | 201/257 | 4291,4309,4343 | 7 |

*(Continued)*

**Table 1.** (Continued)

| Author | Year | Country | Ethnicity | Diagnostic criteria | Sample size (cases/controls) | Polymorphisms | NOS[1] |
|---|---|---|---|---|---|---|---|
| Achouri-Rassa [79] | 2016 | Tunis | European descent | DSM-IV | 85/90 | 1799752 | 9 |
| Fekih-Mrissa [80] | 2017 | Tunis | European descent | DSM-IV, NINCDS-ADRDA | 60/120 | 1799752 | 9 |
| Wang [81] | 2017 | China | Chinese | NINCDS-ADRDA | 113/142 | 4343,1800764 | 7 |
| Li [82] | 2018 | China | Chinese | Chinese Medical Association Criteria | 52/52 | 1799752 | 7 |
| Durmaz [83] | 2019 | Turkey | European descent | DSM-IV | 100/100 | 1799752 | 8 |
| Shu [84] | 2019 | China | Chinese | NINCDS-ADRDA | 149/113 | 1799752 | 7 |

1. Newcastle-Ottawa Scale (NOS) score for each study.

2. National Institute of Neurological and Communicative Disorders and Stroke-Alzheimer's Disease and Related Disorders Association criteria.

3. Revised Diagnostic and Statistical Manual of Mental Disorders-III criteria.

4. Consortium to Establish a Registry for Alzheimer's Disease criteria.

5. Diagnostic and Statistical Manual of Mental Disorders-IV criteria.

### rs1799752 I/D and AD risk

Totally, 57 samples were found dealing with rs1799752 polymorphism and AD risk, comprising 8619 cases and 15718 controls. Since 1 sample just provided allele counts, it was only analyzed in the allelic comparison [74]. In overall analyses, the associations between rs1799752 polymorphism and AD susceptibility were identified under allelic comparison (*I* vs. *D*: OR = 1.091, 95% CI = 1.007–1.181, *p* = 0.032) and dominant model (*II* + *ID* vs. *DD*: OR = 1.131, 95% CI = 1.008–1.270, *p* = 0.036). However, the FDR values were both higher than 0.05, suggesting that the associations were not reliable. Sensitivity analysis by excluding studies not in accordance with HWE obtained similar results (Table 3). When analyses were performed particularly in investigations published in English, no reliable associations were identified either.

In subgroup analysis restricted to late-onset individuals, our investigation indicated significant associations between rs1799752 polymorphism and risk to AD using allelic comparison, additive model (*II* vs. *ID*, *II* vs. *DD*), dominant model and recessive model (*II* vs. *ID* + *DD*). In brief, rs1799752 *I* conferred increased risk to AD compared to the *D* allele with an odds ratio of 1.154 (95% CI = 1.028–1.295, *p* = 0.015, FDR = 0.020). In genotype analysis, the *I* homozygotes showed higher susceptibility for developing AD compared to the *D* homozygotes (OR = 1.308, 95% CI = 1.120–1.528, *p* = 0.001, FDR = 0.003). Moreover, the risk for AD was significantly higher in the *I* homozygotes than in the *D* carriers (OR = 1.272, 95% CI = 1.120–1.444, *p* < 0.001, FDR < 0.001, Fig 2). In cumulative meta-analysis for the *I* homozygotes versus the *D* homozygotes, after only 3 of the 18 cohorts had been studied, the fixed-effect odds

**Table 2. Summary of ACE polymorphisms included in the present meta-analysis.**

| Polymorphisms | No. of cohorts | Total cases | Total controls |
|---|---|---|---|
| rs1799752 I/D | 57 | 8619 | 15718 |
| rs4343 A/G | 23 | 9783 | 16890 |
| rs4291 A/T | 20 | 5973 | 13044 |
| rs1800764 T/C | 20 | 6371 | 6787 |
| rs4309 C/T | 4 | 1187 | 1056 |

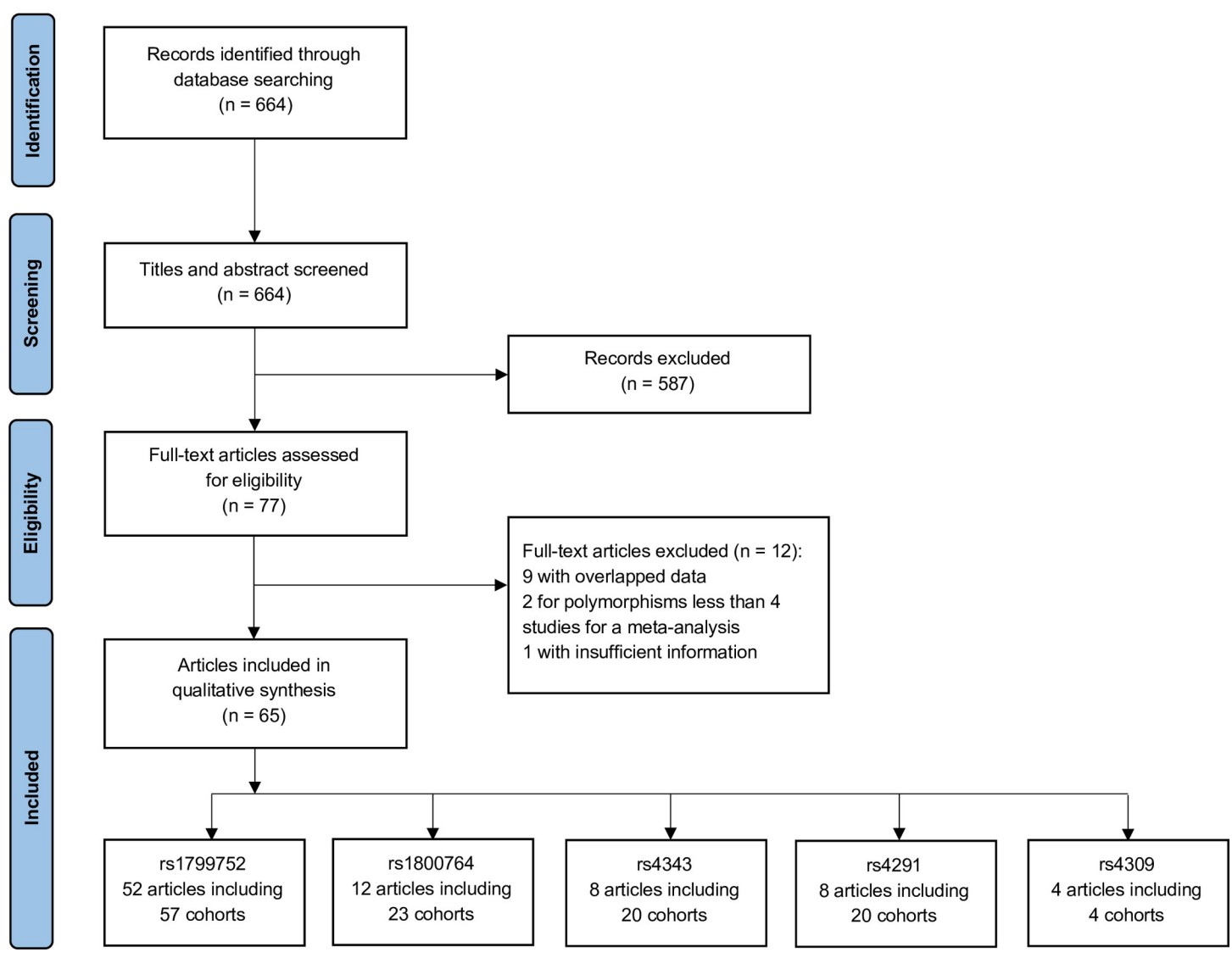

**Fig 1. PRISMA flow diagram for literature search and study selection.**

ratio became larger than 1 and remained so thereafter (Fig 3). However, after excluding 3 cohorts deviated from HWE [30, 31, 55], no positive results could be identified in any genetic model. Subgroup analyses of large sample size cohorts and small sample size cohorts obtained similar results.

We also compared ethnic difference among East Asians (Chinese and Japanese), North Europeans and populations of South European descent (Mediterranean and Middle Eastern). Respectively, 15 studies carried out among East Asians, 16 among North Europeans and 18 in cohorts of South European descent. East Asians had higher *I* allele frequency compared to those of European descent (p < 0.001). In East Asians, the significant associations between rs1799752 and AD susceptibility were revealed using allelic comparison, additive model (*II* vs. *ID*, *II* vs. *DD*), and recessive model. Nevertheless, none of them was robust enough to withstand the FDR adjustment, suggesting the positive results were weak evidence of true associations. In North Europeans, the *I* allele conferred increased risk to AD compared to the *D* allele (OR = 1.096, 95% CI = 1.021–1.178, *p* = 0.012, FDR = 0.039). Meanwhile, the *I* homozygotes

**Table 3. Pooled odds ratios for rs1799752 polymorphism and AD susceptibility.**

| Comparisons | Data | No. of cohorts | Cases/controls | OR | 95% CI | z | p | FDR | I² (%) | p(Q) | Effect model |
|---|---|---|---|---|---|---|---|---|---|---|---|
| I/D | overall | 57 | 8619/15718 | 1.091 | 1.007–1.181 | 2.14 | **0.032** | 0.092 | 67.8 | <0.001 | random |
| | overall in HWE | 47 | 7219/14105 | 1.075 | 0.984–1.174 | 1.60 | 0.109 | 0.194 | 68.5 | <0.001 | random |
| | late onset | 18 | 2664/3322 | 1.154 | 1.028–1.295 | 2.42 | **0.015** | **0.020** | 51.4 | 0.006 | random |
| | late onset in HWE | 15 | 2133/2625 | 1.122 | 0.982–1.281 | 1.69 | 0.091 | 0.137 | 52.6 | 0.009 | random |
| II/ID | overall | 56 | 8584/15633 | 1.067 | 0.989–1.150 | 1.68 | 0.093 | 0.112 | 49.9 | <0.001 | fixed |
| | overall in HWE | 47 | 7219/14105 | 1.054 | 0.969–1.145 | 1.23 | 0.219 | 0.263 | 45.8 | <0.001 | fixed |
| | late onset | 18 | 2664/3322 | 1.214 | 1.060–1.390 | 2.80 | **0.005** | **0.010** | 46.5 | 0.016 | fixed |
| | late onset in HWE | 15 | 2133/2625 | 1.192 | 1.018–1.396 | 2.19 | **0.029** | 0.087 | 48.3 | 0.019 | fixed |
| ID/DD | overall | 56 | 8619/15718 | 1.117 | 0.999–1.250 | 1.93 | 0.053 | 0.092 | 52.8 | <0.001 | random |
| | overall in HWE | 47 | 7219/14105 | 1.094 | 1.012–1.182 | 2.27 | **0.023** | 0.138 | 49.8 | <0.001 | fixed |
| | late onset | 18 | 2664/3322 | 1.098 | 0.966–1.247 | 1.43 | 0.151 | 0.151 | 47.4 | 0.014 | fixed |
| | late onset in HWE | 15 | 2133/2625 | 1.098 | 0.953–1.264 | 1.29 | 0.196 | 0.200 | 10.9 | 0.331 | fixed |
| II/DD | overall | 56 | 8619/15718 | 1.154 | 0.994–1.341 | 1.88 | 0.061 | 0.092 | 60.3 | <0.001 | random |
| | overall in HWE | 47 | 7219/14105 | 1.138 | 0.963–1.344 | 1.52 | 0.129 | 0.194 | 60.9 | <0.001 | random |
| | late onset | 18 | 2664/3322 | 1.308 | 1.120–1.528 | 3.39 | **0.001** | **0.003** | 39.5 | 0.044 | fixed |
| | late onset in HWE | 15 | 2133/2625 | 1.218 | 1.021–1.453 | 2.19 | **0.029** | 0.087 | 33.5 | 0.101 | fixed |
| II+ID/DD | overall | 56 | 8619/15718 | 1.131 | 1.008–1.270 | 2.10 | **0.036** | 0.092 | 60.5 | <0.001 | random |
| | overall in HWE | 47 | 7219/14105 | 1.138 | 0.963–1.344 | 1.52 | 0.129 | 0.194 | 60.9 | <0.001 | random |
| | late onset | 18 | 2664/3322 | 1.156 | 1.026–1.304 | 2.38 | **0.017** | **0.020** | 47.5 | 0.013 | fixed |
| | late onset in HWE | 15 | 2133/2625 | 1.138 | 0.996–1.299 | 1.90 | 0.057 | 0.114 | 20.7 | 0.222 | fixed |
| II/ID+DD | overall | 56 | 8619/15718 | 1.083 | 0.963–1.218 | 1.33 | 0.183 | 0.183 | 57.8 | <0.001 | random |
| | overall in HWE | 47 | 7219/14105 | 1.067 | 0.937–1.214 | 0.97 | 0.330 | 0.330 | 57.3 | <0.001 | random |
| | late onset | 18 | 2664/3322 | 1.272 | 1.120–1.444 | 3.71 | **<0.001** | **<0.001** | 43.8 | 0.024 | fixed |
| | late onset in HWE | 15 | 2133/2625 | 1.162 | 0.924–1.462 | 1.28 | 0.200 | 0.200 | 51.7 | 0.011 | random |

OR: odds ratio; 95% CI: 95% confidence interval; z: test for overall effect; I²: I² value for heterogeneity test.

FDR: adjusted p value using Benjamini-Hochberg (BH) method.

p(Q): p value of the Dersimonian and Laird's Q test for heterogeneity evaluation.

showed higher susceptibility to AD compared with the *D* homozygotes (OR = 1.202, 95% CI = 1.040–1.390, p = 0.013, FDR = 0.039). After excluding those cohorts not in accordance with HWE, the positive associations in North Europeans were more obvious, not only under allelic comparison and homozygotes comparison, but also using additive model (*ID* vs. *DD*: OR = 1.266, 95% CI = 1.045–1.534, p = 0.016, FDR = 0.024) and dominant model (*II* + *ID* vs. *DD*: OR = 1.197, 95% CI = 1.062–1.350, p = 0.003, FDR = 0.018, Fig 4). With regard to populations of South European descent, no significant association was found under all comparisons (Table 4).

Seven studies provided data about rs1799752 polymorphism and *ε2/ε3/ε4* genotypes. With regard to AD susceptibility, no significant association was identified in both APOE *ε4* positive and negative subgroups. However, after exclusion of HWE-deviated samples [27, 51], in the *I* homozygotes and the *D* homozygotes, the presence of APOE *ε4* increased the risk of AD 2.84-fold (95% CI = 1.825–4.418, p < 0.001) and 7.06-fold (95% CI = 3.963–12.571, p < 0.001), respectively.

## rs1800764 T/C and AD risk

Analysis for rs1800764 T/C polymorphism was available on 20 samples, including 6371 cases and 6768 controls. All the above investigations were published in English. In total, the *T*

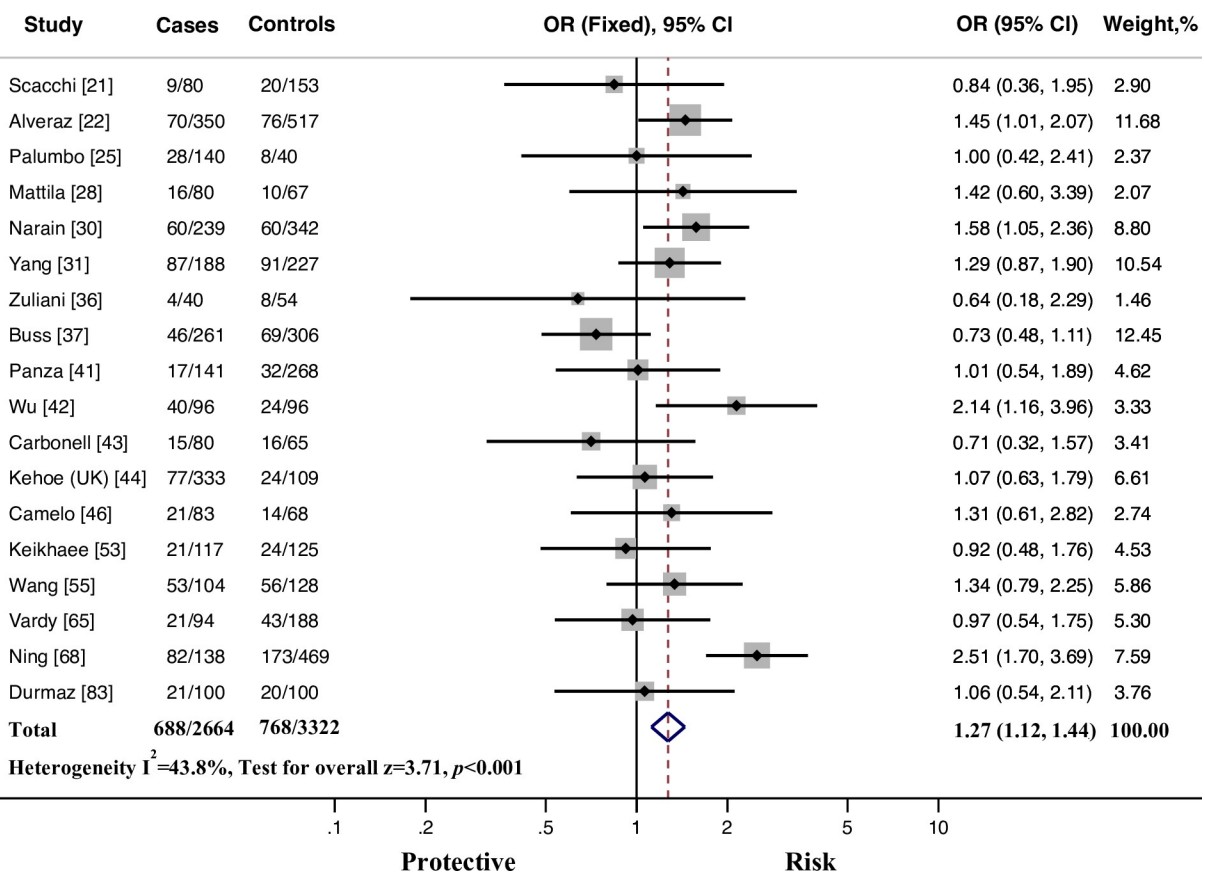

| Study | Cases | Controls | OR (Fixed), 95% CI | OR (95% CI) | Weight,% |
|---|---|---|---|---|---|
| Scacchi [21] | 9/80 | 20/153 | | 0.84 (0.36, 1.95) | 2.90 |
| Alveraz [22] | 70/350 | 76/517 | | 1.45 (1.01, 2.07) | 11.68 |
| Palumbo [25] | 28/140 | 8/40 | | 1.00 (0.42, 2.41) | 2.37 |
| Mattila [28] | 16/80 | 10/67 | | 1.42 (0.60, 3.39) | 2.07 |
| Narain [30] | 60/239 | 60/342 | | 1.58 (1.05, 2.36) | 8.80 |
| Yang [31] | 87/188 | 91/227 | | 1.29 (0.87, 1.90) | 10.54 |
| Zuliani [36] | 4/40 | 8/54 | | 0.64 (0.18, 2.29) | 1.46 |
| Buss [37] | 46/261 | 69/306 | | 0.73 (0.48, 1.11) | 12.45 |
| Panza [41] | 17/141 | 32/268 | | 1.01 (0.54, 1.89) | 4.62 |
| Wu [42] | 40/96 | 24/96 | | 2.14 (1.16, 3.96) | 3.33 |
| Carbonell [43] | 15/80 | 16/65 | | 0.71 (0.32, 1.57) | 3.41 |
| Kehoe (UK) [44] | 77/333 | 24/109 | | 1.07 (0.63, 1.79) | 6.61 |
| Camelo [46] | 21/83 | 14/68 | | 1.31 (0.61, 2.82) | 2.74 |
| Keikhaee [53] | 21/117 | 24/125 | | 0.92 (0.48, 1.76) | 4.53 |
| Wang [55] | 53/104 | 56/128 | | 1.34 (0.79, 2.25) | 5.86 |
| Vardy [65] | 21/94 | 43/188 | | 0.97 (0.54, 1.75) | 5.30 |
| Ning [68] | 82/138 | 173/469 | | 2.51 (1.70, 3.69) | 7.59 |
| Durmaz [83] | 21/100 | 20/100 | | 1.06 (0.54, 2.11) | 3.76 |
| **Total** | **688/2664** | **768/3322** | | **1.27 (1.12, 1.44)** | **100.00** |

**Heterogeneity I²=43.8%, Test for overall z=3.71, *p*<0.001**

.1  .2  .5  1  2  5  10

**Protective**                    **Risk**

**Fig 2. Fixed-effects odds ratio (OR) for the association of ACE rs1799752 *I/D* polymorphism with late-onset AD susceptibility (*II* vs. *ID* +*DD*).** *I* represents the insertion allele, and *D* represents the deletion allele. The size of the gray box is proportional to the weight of the corresponding study. The pooled estimate is displayed as a diamond. Bars, 95% confidence interval (CI).

carriers demonstrated increased risk for developing AD compared with the *C* homozygotes, but the FDR value is insignificant (OR = 1.099, 95% CI = 1.005–1.201, *p* = 0.038, FDR = 0.114). Similar results were found in large sample size studies and after excluding studies not in accordance with HWE (Table 5). Since there was insufficient information to allow subgroup analysis in East Asians or in populations of South European descent, meta-analysis was only performed in all cohorts of European descent. Significant associations were revealed in populations of European descent under allelic comparison (*T* vs. *C*: OR = 1.063, 95% CI = 1.008–1.120, *p* = 0.023, FDR = 0.046) and additive model (*TT* vs. *CC*: OR = 1.136, 95% CI = 1.022–1.262, *p* = 0.018, FDR = 0.046). Furthermore, The *T* carriers conferred increased risk to develop AD compared with the *C* homozygotes (OR = 1.116, 95% CI = 1.018–1.222, *p* = 0.019, FDR = 0.046, Fig 5). No significant association was revealed in all comparisons in late-onset subgroup analysis. Nevertheless, after excluding 2 cohorts not satisfying HWE [71], all the associations became insignificant.

## Other polymorphisms and AD risk

23 studies containing 9783 cases and 16890 controls studied the association between rs4343 A/ G polymorphism and AD risk. 20 studies including 5973 cases and 13044 controls evaluated the correlation of rs4291 A/T polymorphism and AD susceptibility. For rs4309 C/T, 4 studies involving 1187 cases and 1056 controls were included. All of those investigations were

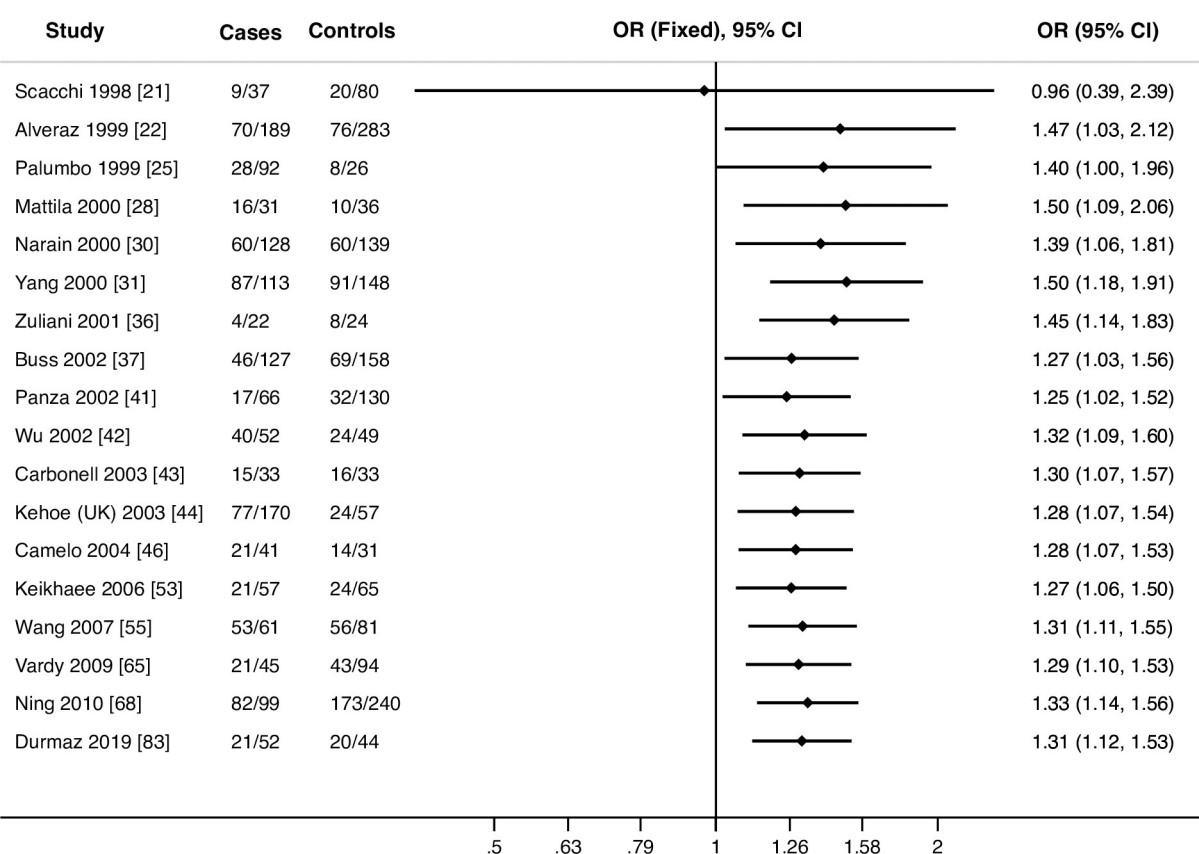

| Study | Cases | Controls | OR (Fixed), 95% CI | OR (95% CI) |
|-------|-------|----------|--------------------|-------------|
| Scacchi 1998 [21] | 9/37 | 20/80 | | 0.96 (0.39, 2.39) |
| Alveraz 1999 [22] | 70/189 | 76/283 | | 1.47 (1.03, 2.12) |
| Palumbo 1999 [25] | 28/92 | 8/26 | | 1.40 (1.00, 1.96) |
| Mattila 2000 [28] | 16/31 | 10/36 | | 1.50 (1.09, 2.06) |
| Narain 2000 [30] | 60/128 | 60/139 | | 1.39 (1.06, 1.81) |
| Yang 2000 [31] | 87/113 | 91/148 | | 1.50 (1.18, 1.91) |
| Zuliani 2001 [36] | 4/22 | 8/24 | | 1.45 (1.14, 1.83) |
| Buss 2002 [37] | 46/127 | 69/158 | | 1.27 (1.03, 1.56) |
| Panza 2002 [41] | 17/66 | 32/130 | | 1.25 (1.02, 1.52) |
| Wu 2002 [42] | 40/52 | 24/49 | | 1.32 (1.09, 1.60) |
| Carbonell 2003 [43] | 15/33 | 16/33 | | 1.30 (1.07, 1.57) |
| Kehoe (UK) 2003 [44] | 77/170 | 24/57 | | 1.28 (1.07, 1.54) |
| Camelo 2004 [46] | 21/41 | 14/31 | | 1.28 (1.07, 1.53) |
| Keikhaee 2006 [53] | 21/57 | 24/65 | | 1.27 (1.06, 1.50) |
| Wang 2007 [55] | 53/61 | 56/81 | | 1.31 (1.11, 1.55) |
| Vardy 2009 [65] | 21/45 | 43/94 | | 1.29 (1.10, 1.53) |
| Ning 2010 [68] | 82/99 | 173/240 | | 1.33 (1.14, 1.56) |
| Durmaz 2019 [83] | 21/52 | 20/44 | | 1.31 (1.12, 1.53) |

.5 .63 .79 1 1.26 1.58 2

**Fig 3. Cumulative meta-analysis of the relation between ACE rs1799752 *I/D* polymorphism and late-onset AD susceptibility (*II* vs *DD*).** Each study was used as an information step. The vertical dotted line is the summary odds ratio. Bars, 95% confidence interval (CI).

published in English. No significant associations were identified in terms of allelic comparison for all these three polymorphisms (rs4343 *A* vs. *G*: OR = 1.002, 95% CI = 0.926–1.084, p = 0.962, FDR = 0.962; rs4291 *A* vs. *T*: OR = 1.025, 95% CI = 0.973–1.080, p = 0.360, FDR = 0.540; rs4309 *C* vs. *T*: OR = 1.072, 95% CI = 0.758–1.517, *p* = 0.694, FDR = 0.790). In further genotype comparison, we still did not reveal any significant association for the three polymorphisms, no matter using additive model, dominant model or recessive model. Subgroup analysis restricted to late-onset individuals, populations of European descent, or large sample size studies for rs4343 A/G and rs4291 A/T polymorphisms obtained similar results.

## Publication bias

Both the Begg-Mazumdar test and the Egger's regression asymmetry test were conducted to evaluate potential publication bias. For all the above polymorphisms, the *p* values of Begg-Mazumdar tests and Egger's tests were greater than 0.05, suggesting no evidence was found for the presence of publication bias.

## Discussion

In this comprehensive meta-analysis about ACE polymorphisms and AD susceptibility on the basis of all available updated studies published in both Chinese and English, 82 cohorts were involved, comprising more than 47000 genotyped cases and controls. Our results demonstrated the significant associations between rs1799752 polymorphism and AD susceptibility in

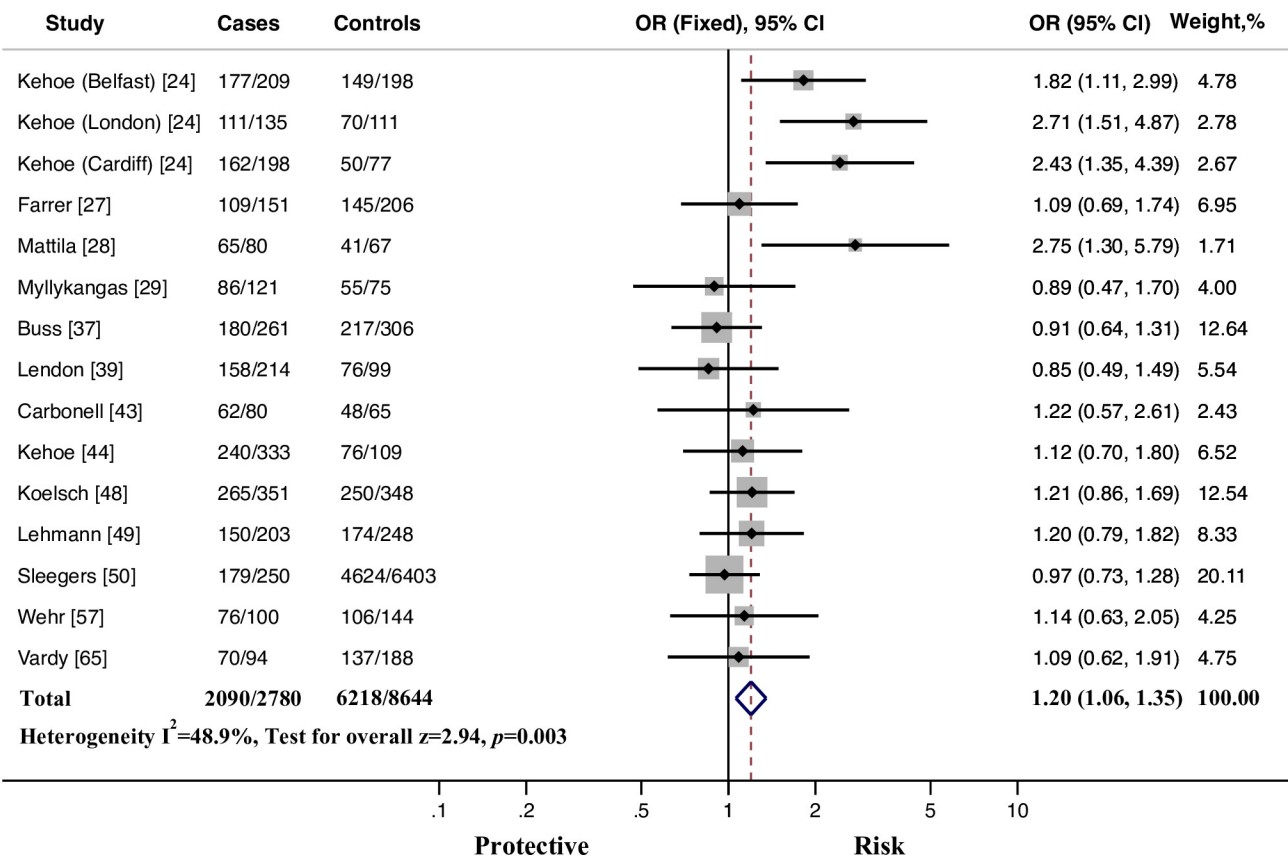

**Fig 4. Fixed-effects odds ratio (OR) for the association of ACE rs1799752 I/D polymorphism AD susceptibility in North Europeans in accordance with HWE (II + ID vs DD).** *I* represents the insertion allele, and *D* represents the deletion allele. The size of the gray box is proportional to the weight of the corresponding study. The pooled estimate is displayed as a diamond. Bars, 95% confidence interval (CI).

North Europeans, but not in East Asians and populations of South European descent, suggesting the ethnic difference of the role that rs1799752 polymorphism played on AD risk. However, our results did not support the associations between rs4343, rs4291 and rs4309 polymorphisms and susceptibility to AD.

Meta-analysis has been considered as a useful tool to achieve more precise estimation of the effect of candidate polymorphism in multifactorial diseases, such as AD. Small sample size, in combination with modest effect, might lead to low statistical power of individual study, which is the most likely explanation for the controversial results of previous investigations. Meta-analysis is one strategy to increase sample size in an attempt to reduce random error which might produce false-positive or false-negative associations. Since deviation from HWE usually means mistyping or selection bias [85], we also excluded those samples not satisfying HWE in controls in sensitivity analysis to increase the accuracy of our results.

For rs1799752 polymorphism, compared to previous meta-analyses performed in 2004 and 2005 [49, 86], we included more than 20 new studies and covered all articles published in Chinese. In overall analyses, before multiple comparison adjustment, our study indicated that the *I* allele conferred increased risk to AD compared with the *D* allele and the *D* homozygotes were at reduced risk of AD compared to the *I* carriers, which were in accordance with results from the two previous meta-analyses. To avoid false positive in multiple comparisons, we applied the widely accepted FDR adjustment in our study [87]. Since no significant association passed the FDR adjustment, we believed that the associations were not robust enough. In

**Table 4. Pooled odds ratios of rs1799752 I/D polymorphism and AD risk by ethnic group.**

**A**

| | East Asians (1848/2400[a], 58.2±11.3%[b]) | | | | | North Europeans (3019/8986, 46.1±3.5%) | | | | | South European descent (all in HWE) (2932/3542, 41.0±5.7%) | | | | |
|---|---|---|---|---|---|---|---|---|---|---|---|---|---|---|---|
| | OR | 95% CI | p | I² | FDR | OR | 95% CI | p | I² | FDR | OR | 95% CI | p | I² | FDR |
| I/D | 1.308 | 1.021–1.675 | **0.034** | 84.8% | 0.068 | 1.096 | 1.021–1.178 | **0.012** | 22.1% | **0.039** | 0.956 | 0.888–1.029 | 0.232 | 45.7% | 0.363 |
| II/ID | 1.330 | 1.001–1.767 | **0.049** | 73.2% | 0.074 | 1.072 | 0.943–1.219 | 0.289 | 41.0% | 0.289 | 0.916 | 0.791–1.061 | 0.242 | 0.0% | 0.363 |
| ID/DD | 1.250 | 0.874–1.786 | 0.221 | 67.4% | 0.221 | 1.209 | 0.989–1.478 | 0.064 | 60.2% | 0.096 | 0.984 | 0.879–1.101 | 0.775 | 24.9% | 0.775 |
| II/DD | 1.676 | 1.040–2.701 | **0.034** | 81.2% | 0.068 | 1.202 | 1.040–1.390 | **0.013** | 23.0% | **0.039** | 0.910 | 0.781–1.061 | 0.228 | 29.0% | 0.363 |
| II+ID/DD | 1.407 | 0.951–2.082 | 0.087 | 76.7% | 0.104 | 1.209 | 1.013–1.444 | **0.035** | 54.0% | 0.070 | 0.963 | 0.866–1.071 | 0.486 | 39.8% | 0.583 |
| II/ID+DD | 1.419 | 1.044–1.928 | **0.025** | 80.3% | 0.068 | 1.114 | 0.987–1.258 | 0.081 | 25.0% | 0.097 | 0.910 | 0.793–1.045 | 0.183 | 13.5% | 0.363 |

**B**

| | East Asians in HWE (1189/1669, 57.7±13.5%) | | | | | North Europeans in HWE (2780/8644, 46.0±3.7%) | | | | | South European descent in HWE (2932/3542, 41.0±5.7%) | | | | |
|---|---|---|---|---|---|---|---|---|---|---|---|---|---|---|---|
| | OR | 95% CI | p | I² | FDR | OR | 95% CI | p | I² | FDR | OR | 95% CI | p | I² | FDR |
| I/D | 1.467 | 1.043–2.062 | **0.028** | 87.3% | 0.056 | 1.102 | 1.022–1.188 | **0.012** | 26.7% | **0.024** | 0.956 | 0.888–1.029 | 0.232 | 45.7% | 0.363 |
| II/ID | 1.518 | 1.053–2.189 | **0.025** | 75.7% | 0.056 | 1.016 | 0.888–1.163 | 0.814 | 25.4% | 0.814 | 0.916 | 0.791–1.061 | 0.242 | 0.0% | 0.363 |
| ID/DD | 1.257 | 0.776–2.037 | 0.352 | 69.9% | 0.352 | 1.266 | 1.045–1.534 | **0.016** | 51.6% | **0.024** | 0.984 | 0.879–1.101 | 0.775 | 24.9% | 0.775 |
| II/DD | 2.002 | 1.035–3.872 | **0.039** | 83.5% | 0.059 | 1.207 | 1.036–1.405 | **0.016** | 28.1% | **0.024** | 0.910 | 0.781–1.061 | 0.228 | 29.0% | 0.363 |
| II+ID/DD | 1.554 | 0.906–2.664 | 0.109 | 79.0% | 0.131 | 1.197 | 1.062–1.350 | **0.003** | 48.9% | **0.018** | 0.963 | 0.866–1.071 | 0.486 | 39.8% | 0.583 |
| II/ID+DD | 1.644 | 1.089–2.482 | **0.018** | 83.4% | 0.056 | 1.077 | 0.948–1.223 | 0.257 | 17.1% | 0.308 | 0.910 | 0.793–1.045 | 0.183 | 13.5% | 0.363 |

[a] cases/controls

[b] I frequencies in controls.

OR: odds ratio; 95% CI: 95% confidence interval; z: test for overall effect; I²: I² value for heterogeneity test.

FDR: adjusted p value using Benjamini-Hochberg (BH) method.

another investigation adopted false-positive report probability (FPRP) to control false-positive findings, the authors obtained similar results as those from our study [88].

In subgroup analysis, significant associations between rs1799752 polymorphism and late-onset AD risk were revealed using allelic comparison, additive model, dominant model and recessive model. However, as well as those identified for rs1800764 in populations of European descent, after excluding studies not satisfying HWE, no positive results could be obtained, indicating the instability of the associations. Thus, in the future, well-designed large sample size studies to provide more forceful evidence for the possible associations are required.

When cohorts were stratified by ethnic background, significant difference was identified among East Asians, North Europeans and populations of South European descent with regard to rs1799752 polymorphism. East Asians had higher I allele frequencies compared to populations of European descent in controls. Furthermore, the most robust and consistent associations between rs1799752 polymorphism and risk to AD were identified particularly in North Europeans. We attributed the difference of results among Europeans and Asians mainly to the difference of genetic background. However, no comparable investigations have been carried out with regard to mechanisms underlying rs1799752 polymorphism and AD development between different ethnics, which still need further research to clarify. Before FDR adjustment, our results of East Asians were in accordance with those of previous meta-analysis performed in Chinese samples [89]. The ethnic difference of the associations between rs1799752 polymorphism and AD risk should be considered in the design of future studies.

One copy of the APOE ε4 allele may increase the risk of AD by 2–6 times [90]. In our study, in the D homozygotes, the presence of APOE ε4 increased the risk of AD around

**Table 5. Pooled odds ratios for rs1800764 polymorphism and AD susceptibility.**

| Comparisons | Data | No. of cohorts | Cases/controls | OR | 95% CI | z | p | FDR | I² (%) | p(Q) | Effect model |
|---|---|---|---|---|---|---|---|---|---|---|---|
| T/C | overall | 20 | 6371/6768 | 1.047 | 0.995–1.102 | 1.75 | 0.080 | 0.120 | 20.1 | 0.205 | fixed |
| | overall in HWE | 18 | 5978/6476 | 1.050 | 0.996–1.107 | 1.83 | 0.068 | 0.160 | 17.2 | 0.138 | fixed |
| | European descent | 18 | 6148/6173 | 1.063 | 1.008–1.120 | 2.27 | **0.023** | **0.046** | 0.0 | 0.494 | fixed |
| | European descent in HWE | 16 | 5755/5881 | 1.067 | 1.011–1.126 | 2.36 | **0.018** | 0.060 | 5.1 | 0.395 | fixed |
| TT/TC | overall | 20 | 6371/6768 | 1.007 | 0.928–1.094 | 0.18 | 0.861 | 0.861 | 2.4 | 0.427 | fixed |
| | overall in HWE | 18 | 5978/6476 | 1.032 | 0.948–1.124 | 0.73 | 0.464 | 0.464 | 0.0 | 0.759 | fixed |
| | European descent | 18 | 6148/6173 | 1.028 | 0.944–1.119 | 0.64 | 0.522 | 0.522 | 0.0 | 0.574 | fixed |
| | European descent in HWE | 16 | 5755/5881 | 1.056 | 0.968–1.153 | 1.23 | 0.219 | 0.219 | 0.0 | 0.928 | fixed |
| TC/CC | overall | 20 | 6371/6768 | 1.097 | 0.998–1.206 | 1.92 | 0.055 | 0.114 | 12.3 | 0.301 | fixed |
| | overall in HWE | 18 | 5978/6476 | 1.073 | 0.974–1.183 | 1.43 | 0.153 | 0.230 | 0.0 | 0.572 | fixed |
| | European descent | 18 | 6148/6173 | 1.105 | 1.003–1.217 | 2.02 | **0.044** | 0.066 | 16.5 | 0.256 | fixed |
| | European descent in HWE | 16 | 5755/5881 | 1.080 | 0.978–1.193 | 1.52 | 0.128 | 0.154 | 0.0 | 0.514 | fixed |
| TT/CC | overall | 20 | 6371/6768 | 1.105 | 0.997–1.224 | 1.91 | 0.057 | 0.114 | 15.0 | 0.267 | fixed |
| | overall in HWE | 18 | 5978/6476 | 1.105 | 0.994–1.228 | 1.85 | 0.065 | 0.160 | 23.0 | 0.182 | fixed |
| | European descent | 18 | 6148/6173 | 1.136 | 1.022–1.262 | 2.37 | **0.018** | **0.046** | 0.0 | 0.512 | fixed |
| | European descent in HWE | 16 | 5755/5881 | 1.137 | 1.120–1.267 | 2.32 | **0.020** | 0.060 | 5.5 | 0.390 | fixed |
| TT+TC/CC | overall | 20 | 6371/6768 | 1.099 | 1.005–1.201 | 1.08 | **0.038** | 0.114 | 16.8 | 0.244 | fixed |
| | overall in HWE | 18 | 5978/6476 | 1.085 | 0.990–1.189 | 1.75 | 0.080 | 0.160 | 14.3 | 0.283 | fixed |
| | European descent | 18 | 6148/6173 | 1.116 | 1.018–1.222 | 2,35 | **0.019** | **0.046** | 12.5 | 0.305 | fixed |
| | European descent in HWE | 16 | 5755/5881 | 1.102 | 1.003–1.210 | 2.02 | **0.043** | 0.086 | 9.6 | 0.344 | fixed |
| TT/TC+CC | overall | 20 | 6371/6768 | 1.034 | 0.957–1.117 | 0.84 | 0.398 | 0.478 | 7.4 | 0.364 | fixed |
| | overall in HWE | 18 | 5978/6476 | 1.052 | 0.971–1.140 | 1.25 | 0.212 | 0.254 | 0.2 | 0.452 | fixed |
| | European descent | 18 | 6148/6173 | 1.058 | 0.977–1.146 | 1.39 | 0.164 | 0.197 | 0.0 | 0.647 | fixed |
| | European descent in HWE | 16 | 5755/5881 | 1.080 | 0.994–1.172 | 1.83 | 0.068 | 0.102 | 0.0 | 0.804 | fixed |

OR: odds ratio; 95% CI: 95% confidence interval; z: test for overall effect; I²: I² value for heterogeneity test.

FDR: adjusted *p* value using Benjamini-Hochberg (BH) method.

*p*(Q): *p* value of the Dersimonian and Laird's Q test for heterogeneity evaluation.

7.06-fold, much higher than that identified in the *I* homozygotes. However, only 5 studies satisfying HWE provided data about rs1799752 polymorphism and APOE *ε4* status, we still need more information to draw safe conclusions about the possible interaction between APOE *ε4* and rs1799752 polymorphism.

In overall analyses for rs1799752 polymorphism, high heterogeneity was identified in different comparisons. After ethnic stratification, high heterogeneity was removed in both North Europeans and cohorts of South European descent, but not in East Asians, suggesting the existence of other confounding factors responsible for heterogeneity. Besides ethnic background, other sources accounting for heterogeneity include differences in sample selection (e.g., age of onset, diagnosis criteria), or in methods (e.g., genotyping methods), or it may be due to interaction with other risk factors (e.g., APOE *ε4* status). When we restricted the study populations to late-onset individuals, heterogeneity among studies reduced. While stratification by APOE *ε4* status nearly removed all the heterogeneity. Our results suggested that age of onset and interaction with APOE *ε4* status also contributed to the high heterogeneity in the analyses of rs1799752 polymorphism.

There are several limitations of our investigation. Firstly, publication bias against reporting negative associations might affect our results. Secondly, no other language article regarding ACE polymorphisms and risk to AD was found besides those in English and Chinese.

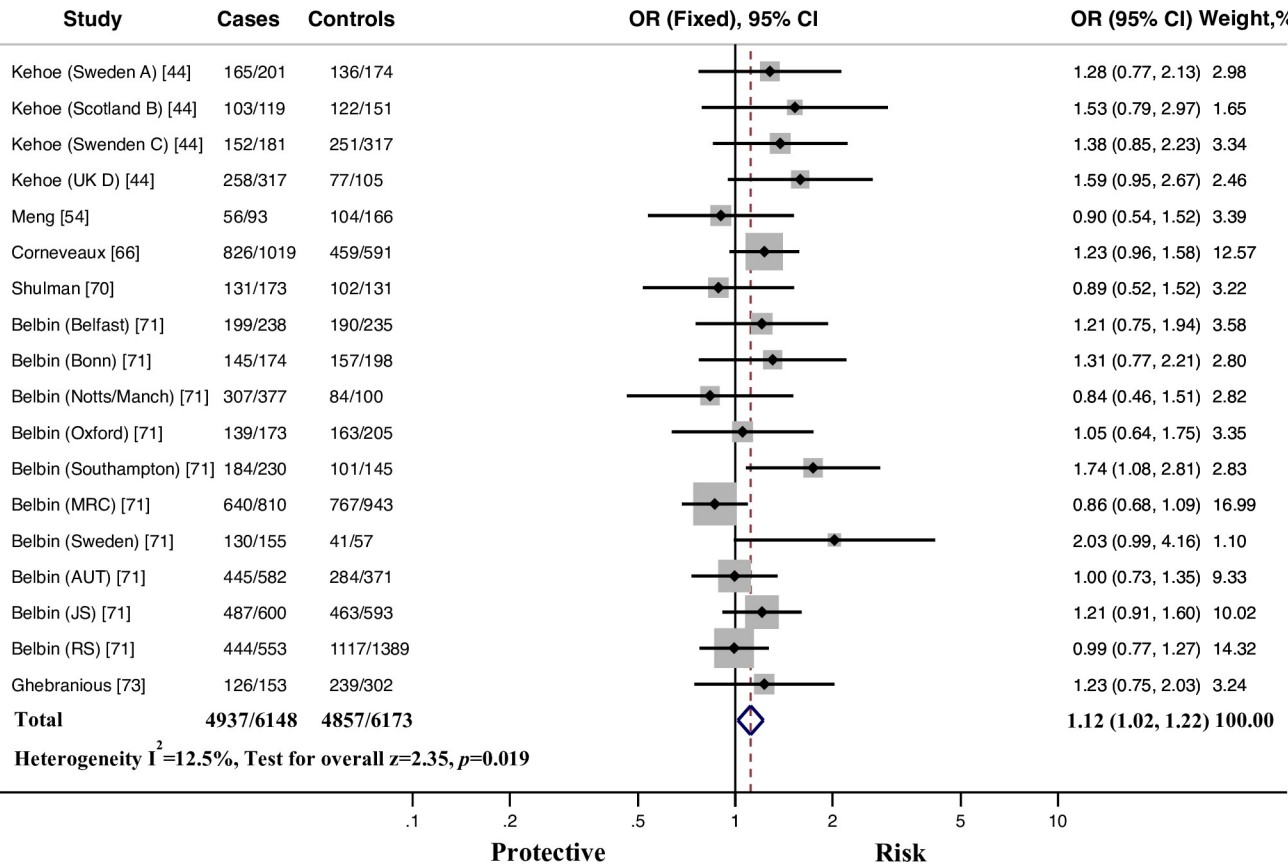

**Fig 5. Fixed-effects odds ratio (OR) for the association of ACE rs1800764 *T*/*C* polymorphism AD susceptibility in populations of European descent (*TT+TC* vs *CC*).** The size of the gray box is proportional to the weight of the corresponding study. The pooled estimate is displayed as a diamond. Bars, 95% confidence interval (CI).

However, some articles cloud be published in journals not on the international journal catalogs, leading to potential language bias. Thirdly, owing to lack of original data, further adjustments by other covariables, such as gender or cardiovascular complications, could not be performed.

## Conclusions

In summary, ACE rs1799752 polymorphism is associated with risk to AD in North Europeans. The relationships between rs1799752 and late-onset AD susceptibility, as well as rs1800764 and AD risk in populations of European descent, still need further studies to illustrate. While rs4343, rs4291 and rs4309 polymorphisms are unlikely to be major factors in AD development in our research.

## Supporting information

**S1 Checklist. PRISMA 2020 checklist.**
(DOCX)

## Author Contributions

**Data curation:** Xiao-Yu Xin, Ze-Hua Lai.

**Methodology:** Kai-Qi Ding.

**Supervision:** Li-Li Zeng.

**Writing – original draft:** Xiao-Yu Xin.

**Writing – review & editing:** Jian-Fang Ma.

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
