## [Decision Letter · Decision Letter 0]

28 Jul 2021

PONE-D-21-18823

Angiotensin-Converting Enzyme Polymorphisms AND Alzheimer’s Disease Susceptibility: An Updated Meta-Analysis

PLOS ONE

Dear Dr. Xin,

Thank you for submitting your manuscript to PLOS ONE. After careful consideration, we feel that it has merit but does not fully meet PLOS ONE’s publication criteria as it currently stands. Therefore, we invite you to submit a revised version of the manuscript that addresses the points raised during the review process.

Please revise carefully as per reviewers' comment. The revision will go back to the same reviewers for their satisfactions.

We look forward to receiving your revised manuscript.

Kind regards,

Zhicheng Lin, Ph.D.

Academic Editor

PLOS ONE

Journal Requirements:

2. Please include the tables to within the manuscript.

3. Please note that according to our submission guidelines (http://journals.plos.org/plosone/s/submission-guidelines), outmoded terms and potentially stigmatizing labels should be changed to more current, acceptable terminology. To this effect, please change 'Caucasian' to 'white' or 'of European descent.

5. Please amend your list of authors on the manuscript to ensure that each author is linked to an affiliation. Authors’ affiliations should reflect the institution where the work was done (if authors moved subsequently, you can also list the new affiliation stating “current affiliation:….” as necessary).

Reviewers' comments:

Reviewer's Responses to Questions

**Comments to the Author**

1. Is the manuscript technically sound, and do the data support the conclusions?

Reviewer #1: Partly

Reviewer #2: Yes

2. Has the statistical analysis been performed appropriately and rigorously? 

Reviewer #1: Yes

Reviewer #2: Yes

3. Have the authors made all data underlying the findings in their manuscript fully available?

Reviewer #1: Yes

Reviewer #2: Yes

4. Is the manuscript presented in an intelligible fashion and written in standard English?

Reviewer #1: Yes

Reviewer #2: Yes

5. Review Comments to the Author

Reviewer #1: This meta analysis by Dr. Xin concluded studies about ACE and AD, aiming to aimed to clarify the effect of ACE polymorphisms on AD risk. The whole research is quite comprehensive and objective, but there are still some shortcomings.Here are some of my suggestions.

1.Some viewpoints need more supporting materials or references, such as “the ε4 allele

accounts for an estimated 45% to 60% of the genetic susceptibility to AD”

2.“Angiotensin converting enzyme (ACE) is an important component of the renin-angiotensin

system (RAS), which mainly acts on promoting the formation of Angiotensin II (Ang II) from

Angiotensin I (Ang I). Recently, many evidences supported that ACE participated in the pathogenesis of AD. As a membrane-bound zinc metalloprotease, ACE played an important role in Aβ degradation. ” should be revised as “Recently, many evidences supported that ACE participated in the pathogenesis of AD. As a membrane-bound zinc metalloprotease, ACE played an important role in Aβ degradation. Angiotensin converting enzyme (ACE) is an important component of the renin-angiotensin system (RAS), which mainly acts on promoting the formation of Angiotensin II (Ang II) from Angiotensin I (Ang I).”. It reads more fluent and logistic.

3.Some sentences were obscure, like “We also stratified the study cohorts, when possible, according to the age of onset, ethnic background and APOE ε4 status.” The relationship between ACE and APOEε4 should be illustrated more.

4.Are there any RCTs (the most convincing type in meta analysis) involved in this research? If yes, pls add one or two sentences in “study election”. If not, pls also explain the reason.

5.In the conclusion, the sentence “While rs4343, rs4291 and rs4309 polymorphisms are unlikely to be major factors in AD development” sounds kind of absolute. It could be better when add “in our/this research” in the end.

Reviewer #2: Xin et al., described their findings in literature meta-analyses of genetic polymorphisms of ACE and indicated that the better associations between rs1799752 polymorphism and risk to AD in North Europeans. But the rs4343, rs4291 and rs4309 polymorphisms are not significantly correlated with AD development. Overall, this paper described well, but still have some problems and questions need further clarification and revision.

Comments:

1. Please identify what is I and D in the Abstract.

2. In the introduction, the authors described “However, the ε4 allele

accounts for an estimated 45% to 60% of the genetic susceptibility to AD, which is neither necessary nor sufficient to cause the disease”. But there are still some references supporting APOE ε4 which increases risk for Alzheimer's disease and is also associated with an earlier age of disease onset. Please clarify.

3. In literature search, the authors indicated that no language restriction. However, normally a better selection criterion is the literature written in English.

4. In study selection, the authors should try to select the cohort with higher numbers of subjects. Because they mentioned in the introduction that “some discrepancies may be related to the small sample size of individual studies”.

5. The authors should try to link the correlation between rs1799752 polymorphism or others and their potential effects on the function of ACE.

6. It is suggested to discuss why North Europeans compared to East Asians carrying rs1799752 polymorphism are more susceptible to AD occurrence.

6. PLOS authors have the option to publish the peer review history of their article (what does this mean?). If published, this will include your full peer review and any attached files.

Reviewer #1: **Yes: **Jingwen Li

Reviewer #2: No

---

## [Author Response · Author response to Decision Letter 0]

30 Sep 2021

Replies to Academic Editor

 We are very grateful for your careful work earnestly. We considered every comment and suggestion of you and made cautious revision accordingly as follows: 

Comment 1: Please ensure that your manuscript meets PLOS ONE's style requirements, including those for file naming.

Response: We have revised our manuscript according to the PLOS ONE style templates to make it meet the style requirements, including those for file naming. 

Comment 2: Please include the tables to within the manuscript.

Response: The tables have been included within the manuscript.

Comment 3: Please change ‘Caucasian’to ‘white’ or ‘of European descent’.

Response: According to your suggestion, ‘of European descent’ or ‘Europeans’ has been used to substitute ‘Caucasian’ in the revised manuscript. 

Comment 4: Please ensure that you provide the correct grant numbers for the awards you received for your study in the ‘Funding Information’ section.

Response: Thank you for your careful checks. We have corrected the grant number in ‘Funding Information’ section according to your positive comment. 

Comment 5: Please amend your list of authors on the manuscript to ensure that each author is linked to an affiliation.

Response: We have amended our list of authors to ensure that each author is linked to an affiliation in the revised version. 

Comment 6: Please include captions for your Supporting Information files at the end of your manuscript, and update any in-text citations to match accordingly.

Response: We have added captions for Supporting Information files at the end of our manuscript and updated in-text citations to match accordingly. 

Replies to Reviewer 1

 We would like to express our sincere thanks to you for your valuable suggestions and comments, which enable us to greatly improve the quality of our manuscript. The revised parts that correspond to the comments are listed as follows:

Comment 1: Some viewpoints need more supporting materials or references, such as “the ε4 allele accounts for an estimated 45% to 60% of the genetic susceptibility to AD”

Response: We totally agree with your suggestion. The statement of “the ε4 allele accounts for an estimated 45% to 60% of the genetic susceptibility to AD” has been corrected. We rewrote this part as “Presence of APOE ε4 allele increases risk of AD with a dose-dependent manner, and might lead to an earlier age of disease onset. The frequency of APOE ε4 in AD patients varied among different ethnics groups, ranging from around 40% to 60%, compared to 20%~25% in controls.” We think this modification help us express our viewpoint more clearly and accurately. In addition, we also added two references to support this viewpoint. (seen in the 2nd paragraph of Introduction)

1. Alex Ward, Sheila Crean, Catherine J Mercaldi, Jenna M Collins, Dylan Boyd, Michael N Cook, et al. Prevalence of apolipoprotein E4 genotype and homozygotes (APOE 4/4) among patients diagnosed with Alzheimer's disease: a systematic review and meta-analysis. Neuroepidemiology. 2012;38(1):1-17. DOI: 10.1159/000334607. PMID: 22179327. (Reference 4 in the manuscript)

2. Verghese PB, Castellano JM, Holtzman DM. Apolipoprotein E in Alzheimer's disease and other neurological disorders. Lancet Neurol. 2011;10(3):241-52. DOI: 10.1016/S1474-4422(10)70325-2. PMID: 21349439. (Reference 5 in the manuscript)

Comment 2: “Angiotensin converting enzyme (ACE) is an important component of the renin-angiotensin system (RAS), which mainly acts on promoting the formation of Angiotensin II (Ang II) from Angiotensin I (Ang I). Recently, many evidences supported that ACE participated in the pathogenesis of AD. As a membrane-bound zinc metalloprotease, ACE played an important role in Aβ degradation.” should be revised as “Recently, many evidences supported that ACE participated in the pathogenesis of AD. As a membrane-bound zinc metalloprotease, ACE played an important role in Aβ degradation. Angiotensin converting enzyme (ACE) is an important component of the renin-angiotensin system (RAS), which mainly acts on promoting the formation of Angiotensin II (Ang II) from Angiotensin I (Ang I).”. 

Response: We appreciate your detailed comment. We have revised this part according to your kind suggestion. Now It reads more fluent and logistic. (seen in the 3rd paragraph of Introduction)

Comment 3: Some sentences were obscure, like “We also stratified the study cohorts, when possible, according to the age of onset, ethnic background and APOE ε4 status.” The relationship between ACE and APOE ε4 should be illustrated more.

Response: We agree with your assessment. We have incorporated your comment in the last paragraph of the introduction part. We added two references to support the possible relationship between ACE and APOE ε4, as well as to clarify why we stratified the study cohorts according to APOE ε4 status: “Since some recent evidence suggested that the presence of APOE ε4 influence the behavioural effects of ACE I/D polymorphism in AD, and the protective effects of ACE inhibitors or angiotensin receptor blockers on cognitive decline correlated with APOE ε4 carrier status, we also performed subgroup analyses according to APOE ε4 carrier status if sufficient data could be obtained”. (seen in the last paragraph of Introduction)

1. Oliveira FF, de Almeida SS, Smith MC, Bertolucci PHF. Behavioural effects of the ACE insertion/deletion polymorphism in Alzheimer's disease depend upon stratification according to APOE-ε4 carrier status. Cogn Neuropsychiatry. 2021;26(4):293-305. DOI: 10.1080/13546805.2021.1931085. PMID: 34034613 (Reference 12 in the manuscript)

2. Ouk M, Wu CY, Rabin JS, Jackson A, Edwards JD, Ramirez J, The use of angiotensin-converting enzyme inhibitors vs. angiotensin receptor blockers and cognitive decline in Alzheimer's disease: the importance of blood-brain barrier penetration and APOE ε4 carrier status. Alzheimers Res Ther. 2021 Feb 11;13(1):43. DOI: 10.1186/s13195-021-00778-8. PMID: 33573702 (Reference 13 in the manuscript)

Comment 4: Are there any RCTs (the most convincing type in meta-analysis) involved in this research? If yes, pls add one or two sentences in “study election”. If not, pls also explain the reason.

Response: You have raised an important concern, and we believe that RCTs are the most convincing type in meta-analysis. However, randomized controlled design might be more suitable for clinical trials rather than gene polymorphism studies of patients and controls. We could not identify any RCTs in electronic database search in the present investigation. We have added illustration to this point as follows in Study selection: “Since all the included studies were gene polymorphism investigations in patients and controls, which were not suitable for randomized controlled design, no randomized controlled trial (RCT) could be identified and involved in our research.” (seen in the 1st paragraph of Study selection in Materials and methods)

Comment 5: In the conclusion, the sentence “While rs4343, rs4291 and rs4309 polymorphisms are unlikely to be major factors in AD development” sounds kind of absolute. It could be better when add “in our/this research” in the end.

Response: We agree with your advice. We have added the words “in our research” in the end of this sentence in the revised manuscript. (seen in Conclusions)

Sincerely yours,

Xiao-Yu Xin  

Replies to Reviewer 2

 Thanks a lot for your generous and detailed comments. Your suggestions provide an important direction for us to revise our paper. We itemized our point-by-point responses to your comments as follows:

Comment 1: Please identify what is I and D in the Abstract.

Response: Thank you for your reminding, we have identified what is I and D in Abstract in the revised version: “the insertion (I) allele conferred increased risk to AD compared to the deletion (D) allele”. (seen in Abstract) 

Comment 2: In the introduction, the authors described “However, the ε4 allele accounts for an estimated 45% to 60% of the genetic susceptibility to AD, which is neither necessary nor sufficient to cause the disease”. But there are still some references supporting APOE ε4 which increases risk for Alzheimer's disease and is also associated with an earlier age of disease onset. Please clarify.

Response: Thank you for pointing out the potential for misunderstanding. We have adjusted the text to be clearer. According to your nice suggestion, we revised this part as follows: “Presence of APOE ε4 allele increases risk of AD with a dose-dependent manner, and might lead to an earlier age of disease onset. The frequency of APOE ε4 in AD patients varied among different ethnic groups, ranging from around 40% to 60%, compared to 20%~25% in controls. Therefore, the presence of ε4 is neither necessary nor sufficient to cause the disease, indicating the participant of other heritable risk factors underlying the development of AD”. We also added two references to support the above statements. (seen in the 2nd paragraph of Introduction)

1. Alex Ward, Sheila Crean, Catherine J Mercaldi, Jenna M Collins, Dylan Boyd, Michael N Cook, et al. Prevalence of apolipoprotein E4 genotype and homozygotes (APOE 4/4) among patients diagnosed with Alzheimer's disease: a systematic review and meta-analysis. Neuroepidemiology. 2012;38(1):1-17. DOI: 10.1159/000334607. PMID: 22179327. (Reference 4 in the manuscript)

2. Verghese PB, Castellano JM, Holtzman DM. Apolipoprotein E in Alzheimer's disease and other neurological disorders. Lancet Neurol. 2011;10(3):241-52. DOI: 10.1016/S1474-4422(10)70325-2. PMID: 21349439. (Reference 5 in the manuscript)

Comment 3: In literature search, the authors indicated that no language restriction. However, normally a better selection criterion is the literature written in English.

Response: You have raised an interesting concern. We believe that high quality investigations usually published in English. However, the strength of meta-analysis depends on combining all available published data to increase statistical power, including those not written in English. Therefore, beside PubMed, Embase and Alzgene, we also searched CNKI (China National Knowledge Infrastructure) for associated studies, which mainly contains literature written in Chinese. Moreover, some high- quality meta-analyses used CNKI for literature search too, such as those published in Lancet series [1,2]. In addition, we used Newcastle-Ottawa Scale (NOS) to assess the methodological quality of each involved study, which has been widely used in meta-analysis (Table 1) [3]. In our analyses for rs1799752, 7 studies were published in Chinese, and others in English. According to your nice suggestion, after broad analysis, we performed analyses particularly in studies written in English. Similar results were obtained after excluding literature in Chinese. We supplied the above results in our revised manuscript: “When analyses were performed particularly in investigations published in English, no reliable associations were identified either.” (seen in the 1st paragraph of rs1799752 I/D and AD risk in Results). For the other several polymorphisms, all the included investigations were written in English. (seen in the 1st paragraph of rs1800764 T/C and AD risk in Results, the 1st paragraph of Other polymorphisms and AD risk in Results)

1. Slee A, Nazareth I, Bondaronek P, Liu Y, Cheng Z, Freemantle N. Pharmacological treatments for generalised anxiety disorder: a systematic review and network meta-analysis. Lancet. 2019;393(10173):768-777. DOI: 10.1016/S0140-6736(18)31793-8. PMID: 30712879

2. Funk AL, Lu Y, Yoshida K, Zhao T, Boucheron P, van Holten J, et al. Efficacy and safety of antiviral prophylaxis during pregnancy to prevent mother-to-child transmission of hepatitis B virus: a systematic review and meta-analysis. Lancet Infect Dis. 2021;21(1):70-84. DOI: 10.1016/S1473-3099(20)30586-7. PMID: 32805200

3. Stang A. Critical evaluation of the Newcastle-Ottawa scale for the assessment of the quality of nonrandomized studies in meta-analyses. Eur J Epidemiol. 2010;25(9):603-5. DOI:10.1007/s10654-010-9491-z. PMID:20652370 (Reference 14 in the manuscript)

Comment 4: In study selection, the authors should try to select the cohort with higher numbers of subjects. Because they mentioned in the introduction that “some discrepancies may be related to the small sample size of individual studies”.

Response: Thank you for pointing out the ambiguous statement in our article. We meant that small sample size, in combination with modest effect, might contribute to low statistical power, which was an important reason responsible for the diversity of previous results. None of individual studies could show more sufficient statistical power than the results of meta-analysis. Because the strength of meta-analysis is the accumulation of all the available published data to increase statistical power, largely addressing the issue of sample size. We have adjusted the text to be clearer as follows: “While meta-analysis is a well-established means to quantitatively synthesize all association data across studies to reduce heterogeneity and identify minor genetic effects, which largely addressing the issue of sample size.” (seen in the 4th paragraph of Introduction). In addition, according to your comment, we compared pooled ORs of large sample size subgroup (more than 300 cases and controls in total) and small sample size subgroup (less than 300 cases and controls in total) when possible, but no significant difference was identified. We also added this part to the revised manuscript. (seen in the last paragraph of Statistical analysis in Materials and methods, the 2nd paragraph of rs1799752 I/D and AD risk in Results, the 1st paragraph of rs1800764 T/C and AD risk in Results, the 1st paragraph of Other polymorphisms and AD risk in Results)

Comment 5: The authors should try to link the correlation between rs1799752 polymorphism or others and their potential effects on the function of ACE.

Response: Thank you for your kind reminding. We have added the potential effects of rs1700752 polymorphism on ACE function in the 4th paragraph of introduction as follows: “The I/D genotype is regarded as a determinant of ACE expression levels in plasma, cells and tissues. Approximately 50% variability in plasma levels of ACE depends on the rs1799752 polymorphism”. We also added 2 references to support the possible correlation. Unfortunately, we found few studies reporting the functions of other polymorphisms. We only know that “rs1800764 and rs4291 located in the regulatory region of ACE gene, while rs4343 in the exotic region.” Though function of the above polymorphisms is outside our investigation, we believe that you have raised a significant direction in future consideration. (seen in the 4th paragraph of Introduction)

1. Cafiero C, Rosapepe F, Palmirotta R, Re A, Ottaiano MP, Benincasa G, et al. Angiotensin System Polymorphisms' in SARS-CoV-2 Positive Patients: Assessment Between Symptomatic and Asymptomatic Patients: A Pilot Study. Pharmgenomics Pers Med. 2021;14:621-629. DOI: 10.2147/ PGPM.S303666. PMID: 34079337. (Reference 9 in the manuscript)

2. Ghafouri-Fard S, Noroozi R, Omrani MD, Branicki W, Pośpiech E, Sayad A, et al. Angiotensin converting enzyme: A review on expression profile and its association with human disorders with special focus on SARS-CoV-2 infection. Vascul Pharmacol. 2020;130:106680. DOI: 10.1016/j.vph.2020. 106680. PMID: 32423553. (Reference 10 in the manuscript)

Comment 6: It is suggested to discuss why North Europeans compared to East Asians carrying rs1799752 polymorphism are more susceptible to AD occurrence.

Response: We think that ethnic genetic background difference is the main reason responsible for the difference of pooled ORs between North Europeans and East Asians. For example, the prevalence of I allele in Asian controls is higher than in European controls not only in our study but also in a recent report [1]. But the underlying reason hasn’t been identified yet, which still need further investigation to clarify. Thanks to your constructive advice, we have added this part in the 5th paragraph of Discussion as follows: “We attributed the difference of results among Europeans and Asians mainly to the difference of genetic background. However, no comparable investigations have been carried out with regard to mechanisms underlying rs1799752 polymorphism and AD development between different ethnics, which still need further research to clarify.”(seen in the 5th paragraph of Discussion)

1. Hatami N, Ahi S, Sadeghinikoo A, Foroughian M, Javdani F, Kalani N, et al. Worldwide ACE (I/D) polymorphism may affect COVID-19 recovery rate: an ecological meta-regression. Endocrine. 2020;68(3):479-484. DOI: 10.1007/s12020-020-02381-7. PMID: 32542429

Best wishes,

Xiao-Yu Xin

---

## [Decision Letter · Decision Letter 1]

11 Nov 2021

Angiotensin-Converting Enzyme Polymorphisms AND Alzheimer’s Disease Susceptibility: An Updated Meta-Analysis

PONE-D-21-18823R1

Dear Dr. Xin,

We’re pleased to inform you that your manuscript has been judged scientifically suitable for publication and will be formally accepted for publication once it meets all outstanding technical requirements.

Kind regards,

Zhicheng Lin, Ph.D.

Academic Editor

PLOS ONE

Additional Editor Comments (optional):

Reviewers' comments:

Reviewer's Responses to Questions

**Comments to the Author**

1. If the authors have adequately addressed your comments raised in a previous round of review and you feel that this manuscript is now acceptable for publication, you may indicate that here to bypass the “Comments to the Author” section, enter your conflict of interest statement in the “Confidential to Editor” section, and submit your "Accept" recommendation.

Reviewer #1: (No Response)

Reviewer #2: All comments have been addressed

2. Is the manuscript technically sound, and do the data support the conclusions?

Reviewer #1: Yes

Reviewer #2: Yes

3. Has the statistical analysis been performed appropriately and rigorously? 

Reviewer #1: Yes

Reviewer #2: Yes

4. Have the authors made all data underlying the findings in their manuscript fully available?

Reviewer #1: Yes

Reviewer #2: Yes

5. Is the manuscript presented in an intelligible fashion and written in standard English?

Reviewer #1: Yes

Reviewer #2: Yes

6. Review Comments to the Author

Reviewer #1: (No Response)

Reviewer #2: (No Response)

7. PLOS authors have the option to publish the peer review history of their article (what does this mean?). If published, this will include your full peer review and any attached files.

Reviewer #1: No

Reviewer #2: **Yes: **Sheng-Fan Wang

---

## [Editor Report · Acceptance letter]

15 Nov 2021

PONE-D-21-18823R1 

Angiotensin-Converting Enzyme Polymorphisms AND Alzheimer’s Disease Susceptibility: An Updated Meta-Analysis 

Dear Dr. Xin:

I'm pleased to inform you that your manuscript has been deemed suitable for publication in PLOS ONE. Congratulations! Your manuscript is now with our production department. 

Kind regards, 

on behalf of

Professor Zhicheng Lin 

Academic Editor

PLOS ONE